# The Effects of Economic Policy Instruments of Diffuse Water Pollution from Agriculture: A Comparative Analysis of China and the UK

Jinpeng Zou [1], Xiaodong Chen [1], Fang Liu [1], Fang Wang [1,*], Mingling Du [1], Bin Wu [2] and Ni Yang [3]

[1] College of Management, Sichuan Agricultural University, Chengdu 611130, China
[2] Business School, Nottingham University, Nottingham NG8 1BB, UK
[3] Division of Food, Nutrition and Dietetics, University of Nottingham, Sutton Bonington Campus, Loughborough LE12 5RD, UK
* Correspondence: wangfangscnd@sicau.edu.cn; Tel.: +86-182-2759-1282

**Abstract:** The world is facing the challenge of increasing grain production and improving the environment, in which the treatment of diffuse water pollution from agriculture is the important content. Traditional administrative means are still unable to solve the problem of market failure and government failure in controlling water pollution. Economic policy instruments have more advantages in improving market economics and reducing the cost of environmental governance and supervision. They have become an important way to solve pollution and promote the transformation of water pollution prevention and control management. This paper puts forward suggestions and countermeasures for improving China's economic policy instruments by systematically sorting out and analyzing the EPIs in China and the UK. Starting from the whole process of agricultural production, China's water quality governance needs to follow three principles to innovate and comprehensively utilize economic policy instruments. A transparent multi-party information sharing and an efficient supervision system are invisible to water quality governance. China also needs to continue to deepen reforms and pilot projects, especially in terms of governance objectives, public welfare funds, water prices, tradable water rights, and emission rights. This paper could also provide a reference for water pollution control in other developing countries.

**Keywords:** agricultural water pollution; economic policy instruments; governance system; system comparison

## 1. Introduction

In recent years, with the improvement of the effect of agricultural point source pollution control, diffuse water pollution from agriculture (DWP), mainly caused by nitrate and sediment, has become the main factor threatening the water ecological environment in various countries [1]. Undoubtedly, controlling DWP is not only a problem of innovation in prevention and control technologies but also a problem of building a system of economic policy instruments (EPIs) [2]. EPIs refer to the policy or the institution that can adjust or influence the behavior of market players by using economic measures, such as price, taxation, credit, investment, micro-stimulus, and macro-economic regulation, to achieve the coordinated development of economic construction and environmental protection. EPIs have obvious benefit incentives, significant cost-effectiveness, and flexibility of controlled objects, so they are widely used in water pollution control around the world [3]. EPIs could be divided into two categories: one focuses on solving the environmental problems through the "visible hand", namely government intervention called Pigou instruments (such as resource tax, pollution tax (fee), subsidy, deposit refund system, etc.); another focuses on solving environmental problems through the "invisible hand", that is the market mechanism itself, called Coase instruments (such as emission trading system) [4]. Both

theoretical research and practical experience at home and abroad have shown that EPIs are the most effective way to internalize the externalities of environmental problems and realize environmental justice [5,6]. The UK has always been faced with the challenge of the DWP; the cost of controlling DWP is about £250 m every year [7]. The UK government has been committed to controlling DWP through effective EPIs [8–10]. Drawing upon the project of the China Scholarship Council, "Beautiful Countryside Construction", this paper comprehensively and deeply researched the UK's EPIs, compared them with current China's situation, and put forward corresponding countermeasures and suggestions for China's governance of DWP.

## 2. The EPIs in the UK

In England, the agri-food industry, which is inseparable from water, contributes £34 billion and employs 150,000 people per year [11]. However, the use of pesticides and chemical fertilizers in agricultural production may have negative impacts on the local water environment, such as excessive hydrochloride and water eutrophication [12]. The UK joined the European Union (EU) in 1973, and its economic policy on controlling DWP is based on the EU's Common Agricultural Policy. The previous directives were largely carried forward despite Brexit in June 2016 and will remain so until further directives from the government. There are three ministries under three devolved governments in the UK to independently manage and cooperate on controlling DWP (as shown in Figure 1): the department for environment, food, and rural affairs, the Scottish environmental protection agency, and the Northern Ireland environment agency are respectively responsible for the affairs of England, Wales, Scotland, and Northern Ireland. These three regions also have some differences in their policies on DWP control. However, they all follow the polluter pays principle, the beneficiary pays principle, and the user pays principle to develop economic policy tools to control water pollution, and they have gained rich experience and remarkable results.

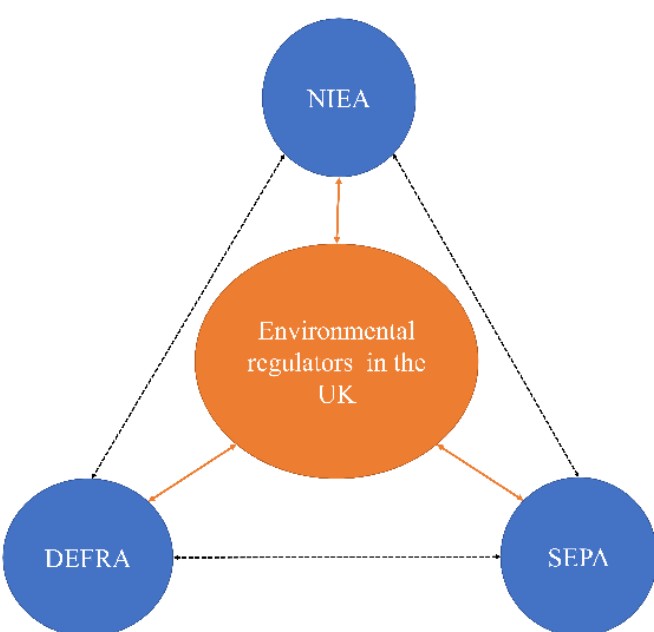

**Figure 1.** Agricultural Environmental Pollution Management System in the UK. DEFRA: department for environment food and rural affairs; SEPA: Scottish environmental protection agency; NIEA: Northern Ireland environment agency.

### 2.1. Financial Subsidy

Farms with greater financial constraints are often among the polluters, especially in the dairy industry [12]. Through financial subsidies, reducing the financial constraints

of producers' pollution control can encourage them to carry out pollution control [13]. Financial subsidies for controlling DWP are mainly divided into two aspects, one of which is production-linked subsidies. It is mainly based on a gross compliance system and a basic payment scheme to provide direct subsidies to agricultural producers for their behavior of controlling DWP. According to statistics, the basic payment scheme 2023 will subsidize nearly 1.8 billion pounds of agricultural subsidies for England [14]. The basic payment scheme has the general requirements for the qualifications of subsidy applicants, among which farmers must have at least five hectares of farming land, they need to plant a variety of crops, and use at least 5% of the land for environmental protection. Meanwhile, they should meet the requirements of the gross compliance system [15]. The gross compliance system includes statutory management requirements and good agricultural and environmental conditions*. The gross compliance system related to controlling DWP is presented in Figure 2a. With the advancement of the UK's agricultural policy reform, production-linked subsidies have been gradually reduced. By 2027, the basic payment scheme will be gradually replaced by the delinked payment, which means that farmers can receive the payments without farming. The second aspect is non-production-linked subsidies, which mainly include financial support for some projects under the rural development program for England. Taking into account the cycle of controlling DWP, the UK government divides subsidies into one-time payments and periodic payments. For example, a grant of up to £250,000 is provided to livestock farmers to cover the cost of the livestock and poultry manure storage equipment [16]. The Countryside Stewardship, with 258 grants, provides periodic funding for farmers to undertake activities to reduce DWP. Catchment-sensitive farming (CSF) not only provides grant funding but also provides free training and consultation to farmers to raise awareness of the spread of agricultural pollution to ensure that no pollutants can be discharged from the fields and farm yards [17]. The latest scheme related to controlling DWP, namely the environmental land management scheme*, mainly includes three plans to improve water quality (Figure 2b). The main change in the future is likely to be that the subsidy will be paid based on the effects of controlling DWP provided by the farmers and other land managers rather than the general requirements of the basic payment scheme. Through the above financial subsidies, UK's agriculture has gradually transformed from the production function to the ecological service function, realizing the coordination of controlling DWP, agricultural production, and farmers' income.

### 2.2. Water Price

In the UK, farms still rely heavily on commercial water, and 86% of farmers tend to use mains water [18]. Commercial water users need to pay the fee in full. The commercial water price adopts the full-cost pricing to ensure cost recovery and a moderate surplus. At the same time, the user's affordability is fully considered. The commercial water price in England is composed of water resource fees and water supply system service fees, which include water supply fees, sewage charges, surface drainage fees, and environmental service fees. Equally, surface water and underground water are also the primary irrigation water sources and are strictly regulated [19]. Their abstraction still requires applying for a permit license in advance, depending on the situation, and paying an application fee and an annual fee. In Scotland and Northern Ireland, where there are abundant water resources, irrigation only needs to apply for permission to abstract the water from the nearby rivers, which is free of charge in most circumstances [20,21]. In England and Wales, for most types of agricultural abstraction, the cost depends on the water consumption applied for each year. If the water consumption exceeds 20 m$^3$ per day, the application fee for the abstraction license and the dependency charge needs to be paid, and these costs will change due to the change of factors, such as pumping volume calculated by modeling, location, water quality, and season [22].

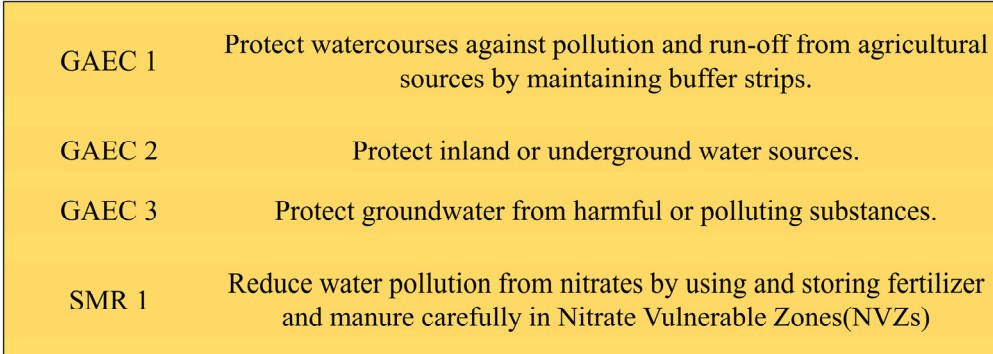

| GAEC 1 | Protect watercourses against pollution and run-off from agricultural sources by maintaining buffer strips. |
|---|---|
| GAEC 2 | Protect inland or underground water sources. |
| GAEC 3 | Protect groundwater from harmful or polluting substances. |
| SMR 1 | Reduce water pollution from nitrates by using and storing fertilizer and manure carefully in Nitrate Vulnerable Zones(NVZs) |

\* SMR : Statutory Management Requirements.
GAEC: Good Agricultural and Environmental Conditions.

(a)

\* ELMS: Environmental land management scheme.

(b)

**Figure 2.** Gross compliance system and newest schemes related to DWP control. (**a**) The gross compliance system related to controlling DWP; (**b**) Three projects in the environmental land management scheme related to DWP control.

### 2.3. Payments for Ecosystem Service

Payments for the ecosystem services can be divided into two types, namely compensation for the ecosystem damage and payment for the ecosystem services. The former follows the polluter pays principle, which is a supplement to binding legal provisions, such as the Environmental Liability Directive (compensation for ecosystem damage caused by accidents) and the environmental impact assessment directive (compensation for ecosystem damage caused by infrastructure projects) from the EU [23]. The latter follows the beneficiary pays principle, which is an economic incentive mechanism to encourage the provision of ecosystem services. There are three broad types of payments for ecosystem service schemes: public payment schemes; private payment schemes; and public-private payment schemes. Payments for ecosystem services also have a wide range of funding sources, covering international, domestic, catchment, and local funding supports. The relevant cases are shown in Table 1.

**Table 1.** The classic cases of payments for ecosystem service in England.

| Case Name | Multiple Participants (B = Buyer, I = Intermediary, S = Seller) | Introduction |
|---|---|---|
| Environmental Stewardship [24] | B = UK government I = Natural England S = farmers and land managers across England | Environmental stewardship is a new agricultural environment strategy, which is based on environmentally-sensitive areas and countryside stewardship schemes, and signed 3 types of multi-year ecological service agreements with land managers: entry-level stewardship; organic entry-level stewardship; higher level stewardship. Farmers can be paid for providing management that reduces water pollution. |

**Table 1.** *Cont.*

| Case Name | Multiple Participants (B = Buyer, I = Intermediary, S = Seller) | Introduction |
|---|---|---|
| Upstream Thinking [25] | B = South West Water I = the River and Wildlife Trusts, National Park Authorities S = farmers and landowners | South West Water invested a total of 20 million pounds in upstream from 2015 to 2020 and provided a total of 1.72 million pounds worth of grants to more than 864 farms to reduce the residues of pesticides, fertilizers, and other pollutants in agricultural water. |
| CSF [26] | B= the department for environment food and rural affairs I = Natural England S = every farm in England | CSF plans to cooperate with farmers in various catchments in England. CSF advisers will provide sellers with one-on-one advice on aquaculture wastewater treatment, water resource management, and pollutant reduction, and helps to obtain funds. The project can better protect water resources and soil management. |
| Wessex Water catchment [27] | B= Wessex Water I= Wessex Water S= farmers in the specific catchments | Wessex Water works with farmers in 15 specific catchment areas to minimize buildup of nitrates, phosphates, agrochemicals and sediment pollution in water by providing advice and funding for cleaner production practices by farmers. |

*2.4. Tradeable Water Right*

In the UK, the whole or part of the tradeable water right among water abstraction license holders can be transferred permanently or temporarily. In general, water trade in the UK is very small; there were only around 50 deals in England and Wales between 2003 and 2011 [28]. The reason for the above phenomenon may be that the water market is immature, and the transaction process may be very bureaucratic and long (mostly more than half a year). To effectively promote the water rights trade and protect water resources, the environment agency has recently planned the potential trading area of water available for each river basin district, which can be divided into water available for licensing, restricted water available for licensing, and not available for licensing [29]. While England is making great efforts to build a tradeable water right network system to improve the efficiency of water trade with two mainstream methods. In the improved pair-wise trading system, a seller needs to find a buyer first, and then the transaction should be approved by the environment agency (Figure 3a). The central pool method requires buyers and sellers not to trade directly, but to build a bridge through the management agency to suggest transactions, so this method is more flexible and is faster (Figure 3b). However, there are still many limitations in the present tradeable water right, such as a lack of market, complicated transactions, and an inability to fully reflect the transaction value. Passive barriers include the lack of a visible market, inability to see the value of transactions, lack of understanding of the transaction process, hoarding funds for future uncertainty, and the existence of transaction substitutes. In addition, the transaction of groundwater rights is more complicated, with more time and application fees [30].

*2.5. Ecological Tax*

Tax policy exists to influence the behavior of those responsible for water pollution by charging mandatory fees, thereby encouraging society to carry out cleaner production and reduce emissions [31]. Department for the environment, food, and rural affairs imposes an ecological tax on pollutants discharged in water bodies, but its share accounts for no more than 2.5% of the country's GDP from 2006 to 2015 [32]. Although there is no taxation on DWP, there are many types of taxes on carbon emissions, such as climate change levies, carbon reduction commitments, and emissions trading schemes.

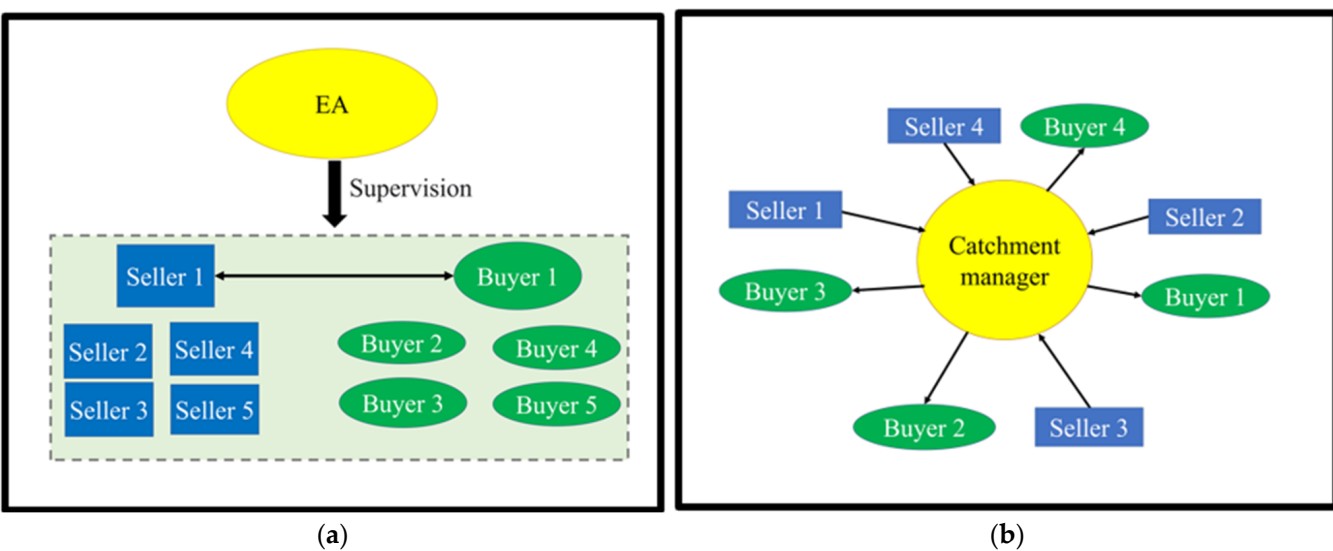

**Figure 3.** The current trading system in the UK [28]. (**a**) The model of improved pair-wise trading; (**b**) The model of the central pool method.

### 2.6. Sustainable Procurement Strategy (SPS)

Governments often view SPS as a useful tool for promoting environmental reform. For example, bids for government contracts may depend on companies promising to meet certain environmental standards. The UK government's sustainable procurement needs to implement the greening government commitments; that is to say, governments' SPS should consider the benefits of coordination of multi-party among organizations, society, and the environment. HM Revenue and Customs are fully committed to the conservation of natural resources and the prevention of environmental pollution. It is UK's policy to conduct business with due regard to environmental needs. HM Revenue and Customs are also asking that each department will set internal targets and continue to reduce energy and water pollution [33]. In addition to the government, large enterprises in the UK have also joined the SPS. Thames Water ensures that any trade in the water agrees to the need to achieve good ecological status or potential with a third-party supplier to identify and pass a strategic environmental assessment to minimize environmental impact, taking into account the socio-economic impacts [34].

### 2.7. Deposit Return Scheme

The UK government planned to introduce this scheme for drinks containers to incentivize people to recycle plastic and glass. The Scottish will collect 90% of certain cans and bottles for recycling after 2023 and help alleviate agricultural plastic pollution [35]. Under this scheme, consumers are required to pay a 20 pennies deposit per bottle when they buy drinks, which they can get back when they return the empty bottle.

### 2.8. Public Benefit Fund

Some measures to mitigate DWP may be adopted by farmers out of an altruistic concern for environmental quality, but for most measures, the motivation for adoption is the cost or time savings. Therefore, it is very important for farmers to obtain scientific and effective guidance. In England, under the advocacy of the government, farmers are active in industry-led voluntary initiatives to implement environmental protection measures [12]. Farmer's volunteering is also facilitated by advice and technical assistance from non-governmental organizations. Most of these non-governmental organizations are registered charities, including river trusts, wildlife trusts, and other farm advisory groups, with funding from the government (UK or EU) and private sources. They often seek to develop and encourage farmers to adopt "win–win" solutions for management

improvement, cost savings, and environmental protection. For example, CSF and the case of Wessex Water work with farmers to improve water quality [36].

### 2.9. Environmental Trade

The UK has already used its trade policy to reduce trade barriers in environmental goods and services. It introduced new global tariffs in early 2021 and removed tariffs on more than 100 environmental goods and services, which is worth more than £2.1 billion; it also lowered tariffs on another 104 environmental goods and services to promote resource reuse and develop the circular economy. The traffic policy has boosted the UK's domestic innovation and exports of environmental goods and services, with seven key export markets for water and waste management products and services [37].

### 2.10. Green Financing

The UK's green financing scheme was launched in September 2021. The funds raised by the UK's green bonds and retail green savings bonds will help finance government spending to tackle climate change, biodiversity loss, and other pressing environmental challenges. A total of 16.4 billion pounds was raised in 2021–2022. These funds are allocated to six categories of expenditure. The funding for DWP control is mainly based on the agri-environment schemes in living and natural resources. A total of 249 million pounds was invested in agri-environmental protection. These two bonds were successfully recognized by peers [38].

### 2.11. Environmental Liability Insurance (ELI)

ELI is a kind of liability insurance with the object of compensation for environmental tort damages or governance responsibilities that the insured should bear due to environmental pollution. The insurance money is mainly used to pay the insured person's liability for damages caused by environmental pollution, and some cases also include the cost of cleaning up the environmental pollution. ELI in the UK was developed in the 1960s because of the pollution caused by industrialization. One of the most typical incidents is the famous "London smog incident" [39]. The amount of compensation caused by environmental pollution is often too huge for ordinary enterprises to bear. As a result, the government will eventually pay the bill, resulting in a secondary violation of public interest. It is for the above reasons that environmental pollution liability insurance began to appear and develop in the UK. The behavior of environmental pollution cannot be regarded as subjective and intentional. The policyholder needs to prevent and mitigate foreseeable environmental pollution within a reasonable range; otherwise, the insurance company has the right to refuse compensation. Most UK ELIs belong to voluntary insurance, supplemented by compulsory insurance.

## 3. The EPIs in China

China mainly conducts DWP control through a multi-departmental coordination mechanism, mainly the ministry of ecology and environment, the ministry of agriculture and rural affairs, the ministry of finance, and the national development and reform commission.

### 3.1. Financial Subsidy

China mainly subsidizes DWP control through project support and financial subsidies. The subsidy projects related to DWP control are shown in Table 2. Most of the Chinese government's subsidies to farmers are one-time post-subsidies and rewards from various departments.

**Table 2.** Subsidies related to DWP control in China.

| The Name of Subsidies | Subsidy Content | Subsidy Object | Regulator | Subsidy Method |
|---|---|---|---|---|
| Soil testing formula fertilization subsidy | Subsidies for soil testing, formula, fertilizer distribution, and other links and project management fees | Agricultural technology extension agencies undertaking soil testing and formula fertilization tasks and enterprises that process formula fertilizers according to formula; farmers | Ministry of Agriculture and Rural Affairs; Ministry of Finance | Reward; post-subsidy |
| Pilot project of comprehensive utilization of livestock and poultry manure | Subsidizing the infrastructure construction of the collection, storage, treatment, and utilization of livestock and poultry manure | Large-scale farms, social service organization for centralized treatment of livestock and poultry manure | Ministry of Ecology and Environment; Ministry of Agriculture and Rural Affairs | Reward; post-subsidy |
| Comprehensive control of DWP in key watersheds | Funding to control DWP in water critical and environmentally sensitive areas | A batch of key typical agricultural small watersheds across the country | Ministry of Agriculture and Rural Affairs; National Development and Reform commission | Reward; post-subsidy; social capital |
| Demonstration subsidies for low-toxicity and low-residue pesticides | Guaranteeing the quality and safety of agricultural products and agricultural ecological safety from the source, and reducing DWP | Farmers; landowners | Ministry of Agriculture and Rural affairs | Reward; post-subsidy |
| Toilet revolution financial incentives | Renovation of household toilets and supporting construction of public facilities | Villages | Ministry of Agriculture and Rural Affairs; Ministry of Finance | Reward; post-subsidy |
| Rural environmental infrastructure construction projects | Accelerating the promotion of domestic sewage treatment in rural areas | Pilot villages and counties | Ministry of Agriculture and Rural affairs | Direct payment |
| Rural smelly water treatment demonstration project | Rural domestic sewage treatment; rural toilet manure treatment; livestock and poultry manure treatment; aquaculture pollution prevention and control; planting industry pollution | Cities | Ministry of Ecology and Environment | Reward, post-subsidy |
| Regulations on the prevention and control of pollution in drinking water source protection areas | Setting up of drinking water source protection area and prevention and control of pollution | Pollution prevention and control management of drinking water surface water sources and groundwater sources of all centralized water supply in the country | Ministry of Ecology and Environment | Reward, post-subsidy |
| Groundwater environmental protection and pollution remediation | Classified control of groundwater pollution | Local governments | Ministry of Ecology and Environment | Reward |

*3.2. Water Price*

Until the 1960s, China's irrigation water was still free, although the Chinese government has continued to reform agricultural water prices for nearly 40 years, trying to use

the role of price levers to make water prices reach a reasonable level to reduce the burden on farmers and save water. However, the goal of agricultural water prices reaching the cost has not been achieved yet [40]. China's agricultural water price includes water resources fees and water supply price fees for farmland water conservancy projects. Since water supply agencies have not fully covered water resources fees and water supply price fees in accordance with market mechanisms, most water supply agencies in the country are in deficit [41]. In order to ensure their operation, the government provides large financial subsidies every year, causing the cost of water price subsidies to be too high. In addition, other agricultural subsidies issued by governments in some regions are directly used to subsidize water fees, making farmers less sensitive to the price of agricultural water. On 21 January 2016, the general office of the state council issued and implemented the "Opinions on promoting the comprehensive reform of agricultural water price", which proposed to establish and improve the formation mechanism of agricultural water price, including setting agricultural water price by consumption and purpose [42]. At present, this scheme is being implemented nationwide, but the speed of the advancement in various regions is uneven, and there is still a long way to go before the goal is fully realized.

### 3.3. Payments for Ecosystem Service

In 2007, the ministry of ecology and environment followed the polluter pays principle and beneficiary pays principle when establishing the pilot of the payments [43]. Since the end of the 20th century, China has approved six forestry payment plans, including natural forest protection projects, forest ecological benefit compensation projects, conversion of farmland to forest projects, three north shelterbelt construction projects, Beijing–Tianjin sandstorm source control projects, etc. Among them, the project of returning farmland to forests is one of the most extensive and effective payments for the ecosystem service in China [44]. In recent years, China has explored the establishment of a horizontal ecological compensation mechanism. China has approved the horizontal ecological compensation mechanism for the Yellow river and the Yangtze river basins across provinces after 2020 [45,46]. However, the current payments for the ecosystem service still have problems, such as lack of legislation, insufficient overall coordination, difficulty in realizing the value of aquatic ecological products, and difficulty in implementing a market-oriented diversified compensation mechanism.

### 3.4. Tradeable Water Right

China's initial attempt at the tradeable water right began with the circulation of water tickets among farmers in Zhangye, Gansu. In July 2014, China launched the pilots of tradeable water right in seven provinces, including Ningxia, Jiangxi, Hubei, Inner Mongolia, Henan, Gansu, and Guangdong [47]. On 28 June 2016, the China Water Exchange officially opened for operation. In 2020, the total transaction scale was 362 million yuan, of which regional water rights and water abstraction rights transactions accounted for 99.89%, but the transaction scale of irrigation water users was only 384,100 yuan [48]. In terms of transaction prices, the average transaction price of regional water rights and water abstraction rights in 2020 was 1.23 yuan/cubic meter, and the highest price was 3.85 yuan/cubic meter, which was much higher than the highest transaction price of 0.6 yuan/cubic meter in 2019. The average transaction price of irrigation user transactions is 0.08 yuan/cubic meter, and the highest price is 2.25 yuan/cubic meter, which is also much higher than the 0.66 yuan/cubic meter in 2019 [48]. Judging from the transactions in 2020, the current price of water right transactions is generally too low. Compared with agricultural water consumption, the trading scale of agricultural water rights is too small, and it still cannot really play a role in improving the allocation of agricultural water resources.

### 3.5. Emission Right

The most important progress in China's emission trading market is the "Regulations on the Management of Pollution Discharge Permits", officially promulgated in 2021. In addi-

tion, the promulgation of 13 national standards in "Technical Guidelines for Self-Monitoring of Pollutant Discharging Units" has laid a solid foundation for the institutionalized development of the emission trading database [49]. From the perspective of trading objects, there are 12 kinds of objects in the national emission trading market, mainly including four binding indicators, namely sulfur dioxide, chemical oxygen demand, ammonia nitrogen, and nitrogen oxides. As of the end of 2021, the transaction value of the above four binding indicators across the country was about 1.9 billion dollars, accounting for about 98% of the total transaction [50]. In addition, each region also increased the trading indicators according to their actual situation. For example, Zhejiang carried out the trade of total phosphorus and volatile organic compounds, and Guangdong carried out the trade of volatile organic compounds; the total nitrogen and phosphorus could be traded in Gansu. Hunan pays attention to heavy metals, such as lead, cadmium, and arsenic. However, China's emission trading is limited to the establishment of a separate trading system at the provincial and municipal levels, resulting in a limited market size and emission reduction effects. Water pollutants usually have the characteristics of diffusion and mobility. Joint management and control of pollutants across regions through regional cooperation can more effectively play the role of the market mechanism in promoting emission reduction and improving the effectiveness of pollutant control. The government-led cap control and emission trading allocation mechanism are not perfect. At present, the paid allocation and pricing methods of various regions are not unified and standardized. There is an auction-based approach to market-based primary distribution, which makes it difficult to establish a real market mechanism for emission trading and truly reflects the value of emission rights. As a result, the participation of emission control companies is inactive, and it is difficult to effectively stimulate market activity.

### 3.6. Ecological Tax (Fee)

Following the polluter pays principle, China started the pilot project of sewage charges in 1979, but in actual implementation, there is a problem of comprehensive supervision by law enforcement agencies [51]. To protect the ecological environment with a strict legal system, the charging system was changed to the ecological tax in 2018. Large-scale livestock and poultry farms with sewage discharge outlets need to apply for sewage discharge permits and pay taxes. However, scattering-raising households, large-scale farms that can comprehensively utilize manure, and farms that do not directly discharge pollutants into the environment can be exempted from the tax. The collection of taxes is very effective in controlling point source pollution, but it may be somewhat helpless for DWP control caused by the farmers' planting and free-range livestock.

### 3.7. ELI

In order to strengthen environmental protection, the state has issued many policies related to ELI in recent years. In January 2013, the Guiding Opinions on the Pilot of compulsory ELI was released, proposing to pilot ELI nationwide. In May 2014, the new Environmental Protection Law was promulgated and implemented to encourage high-risk enterprises to buy ELI. In May 2018, the measures for the administration of compulsory ELI stipulated that enterprises prone to environmental pollution should take out compulsory ELI. By December 2020, China has carried out the pilot project of compulsory ELI across the mainland [52]. However, most of these insurances are concentrated in high-risk industries involving heavy metals, petrochemical, hazardous chemicals, hazardous waste disposal, and other industries. A sound ELI system has not yet been formed for agriculture in China.

### 3.8. Green Financing

The Chinese government has adopted a series of policies to encourage the use of green bonds to promote the financing of environmental protection projects, which greatly encourages institutions and investors to actively participate in the green bond market. By the end of 2020, China had issued about 1.2 trillion yuan of green bonds, ranking second

after the United States in the world [53]. The directory of green bond support projects (2021 version) made it clear that bond financing will be carried out for controlling DWP in China, such as manufacturing of water pollution control equipment, good water body protection, groundwater environment prevention, water environment treatment in key river basins and sea areas, prevention and control of DWP in forestry and grass industry, and prevention and control of DWP in agriculture, forestry, and grass industry [54].

*3.9. SPS*

In 2007, the Chinese government's green procurement system stipulated that state agencies, institutions, and organizations shall not purchase products that endanger the environment and human health with financial funds. The "Government Procurement Category Catalog" (2022) provides very detailed services for controlling DWP, such as sewage treatment and its recycling services, urban waters treatment services, marine waters pollution treatment services, river and lake treatment services, reservoir pollution treatment services, and groundwater pollution control services, showing the government's emphasis on procurement services for controlling DWP [55]. A typical case is a public–private partnership project of livestock and poultry manure treatment in Sichuan. Sichuan takes the agricultural supervision department as the project sponsor. Based on the distribution rights of comprehensive utilization products, such as biogas residue and biogas slurry, the government publicly selects partners through procurement according to law and subsidizes the transportation cost of biogas fertilizer and transportation and storage facilities and equipment with financial funds, eventually returning the biogas manure to the other places lacking organic fertilizer.

## 4. Discussion: System Comparison

Through the above-mentioned analysis of the EPIs of China and the UK, we can find that the EPIs of China and the UK to control DWP are similar, and both of them follow the beneficiary pays principle, polluter pays principle, and user pays principle. Significant achievements have been made in green finance and trade, and at the same time, they face the difficult problem of water rights governance. There are obvious differences in EPIs in the actual operation process between the two countries. An analysis of the causes of the differences will help China build a sound EPI system.

*4.1. Institutional Density*

There are many regulatory enforcement departments in China, and it is hard to say that the responsibilities of each department have no conflicts of interest. Most of China's water resources protection laws and regulations are instructive and implemented by local governments. The rights and responsibilities of the local governments are not clearly defined, resulting in inactive and ineffective implementation by the local authorities [56]. The overlapping responsibilities of DWP supervision are distributed among different departments of the Chinese government, which may easily lead to excessive regulation or a regulatory vacuum phenomenon. Besides, restrictions on data sharing across departments have been an important factor limiting regulatory efficiency [57,58]. In the UK, the environment agency is responsible for DWP control, while water regulation and water safety are managed by the other two departments. It can be said that the division of responsibilities and obligations of the British management department is very clear.

*4.2. Governance Model*

The UK is a model of giving priority to source governance and full control. For example, in CSF, the government directly cooperates with farmers to promote cleaner production of farmers to reduce DWP. Pollutants are not allowed to be discharged from the farm. The government provides a variety of subsidies to farmers with clean production to encourage their green production behavior. The subsidies involve the entire stage of farmers' production. At the same time, the government also encourages farmers to

participate in ecosystem service in order to obtain financial returns. However, China pays attention to the in-processing and post-processing controls. For example, in Table 2, there are relatively few subsidy policies related to cooperation with farmers to protect water resources, and most of them are command-type policy instructions.

### 4.3. Subsidy Target

British subsidies focus on ecological and environmental benefits, and the application is open and transparent. The delinked payments will focus on the results that farmers achieve for environmental protection. For farmers' subsidies, the application conditions, subsidy standards, and application process are all detailed on the government website. If farmers fail to meet the requirements of the agreement, their payments will be reduced or even canceled. The entire application is open and transparent, and the subsidy amount and efficiency for farmers are relatively high. However, China's agricultural subsidies mainly focus on agricultural production, and even though most of the agricultural environmental subsidies are one-time ones, they are not paid directly to the farmers but are used by the government to invest in environmental projects.

### 4.4. Public Benefit Fund

The UK vigorously develops public benefit funds, such as Canal and River Trust, which protects more than 2000 miles of historical waterway relics in the UK by accepting donations from DEFRA and private individuals [57]. Some other charitable funds include The Woodland Trust and the Wildlife Trusts, which are also actively involved in the payments for the ecosystem service projects in the UK and have made positive contributions in participating in the maintenance of biodiversity and environmental protection. At present, China's environmental protection law does not provide for corresponding environmental trusts, and the Trust Law only vaguely states that the state encourages the development of environmental trusts. By July 2021, there have only been 18 environmental public trusts in China, with a property scale of 37,804,000 yuan [59], illustrating that the development of environmental trusts in China is lagging behind.

### 4.5. Water Price

The UK sets water prices based on cost and fully considers the affordability of the users so that cost pricing can be achieved. At present, the price of irrigation water in China is extremely low, and the reform of agricultural water prices is still very difficult. Because the voice of canceling the agricultural water fee has always existed, especially in southern China due to relatively abundant water resources, it is even more difficult to collect agricultural water fees. Therefore, it is necessary to learn from the practice of the United Kingdom to charge agricultural water fees according to the purpose of use, season, and region and give full play to the role of the Chinese farmers' water association, to gradually transition to charging water resource users, and improve the use of water resources efficiency.

### 4.6. Common Challenge

Both China and the UK have problems, such as unclear cross-regional transaction procedures within the river basins, especially the unclear legal status of the tradeable water rights and the emission rights trading, the lack of national technical regulations, and the lack of vitality in the secondary market. It restricts the development of tradeable water emission rights and their credit. Therefore, the two countries can explore the establishment of a special trading framework within each river catchment and gradually improve the relevant laws and regulations on the trading of tradeable water rights and emission rights trading, in particular, providing effective legal protection for farmers' water rights and ecological water rights, which is still an institutional issue that has yet to be resolved.

## 5. Discussion

Comparative research can help researchers analyze, explain and summarize research objectives from comparison [60]. This paper aims to help China, and even developing countries carry out water governance by comprehensively comparing the similarities and differences between China and Britain in the implementation of EPIs. A good DWP supervision system is the key to DWP governance [61]. China is not unaware of a series of problems exposed by the high institutional density. In order to break the deadlock of the "multi-department division and governance" and achieve multi-department coordinated governance of DWP, China established the river chief system in 2016, which is a hybrid authority system designed with the "organizational authority and hierarchical authority". Its realization is based on the "top–down" strict administrative organization structure, which may further aggravate the imbalance between the elements of cross-departmental collaborative governance, and is not conducive to building a good cross-departmental collaborative DWP governance [62]. In fact, if the government adopts the government-led or market-led DWP governance model alone, it will not fully release the potential of EPIs and will bring huge financial pressure to the central government or provincial governments [63,64]. In developed countries, market regulation plays an increasingly important role in ecological compensation, water price, tradable water rights, emission rights, etc.

In addition to the current "integrated management model" represented by the UK, there are river basin committees, water bureaus, or water companies on larger rivers that manage the water resources and water conservancy projects of the river basin in a unified manner. There is also the "centralization–decentralization" model represented by the US, the unified regulatory laws and regulations, and regulatory standards for various departments and regions to manage water resources and the water environment separately according to the division of responsibilities and regions. France also implements "centralization–decentralization" management. Each river basin has a river basin committee and a water council. The former represents the local government rather than the central government and aims to promote the various agencies in the basin to fulfill their roles and responsibilities, while the latter is in the implementation when making decisions; the river basin committee answers to the central government and engages in various specific technical works. These models are characterized by the joint participation of exclusive watershed institutions, governments, village collectives, and residents who own land in the watershed, involving self-governance, joint governance, and hierarchical governance [62,65,66]. The purely "bottom–up" initiatives driven by farmers' voluntary efforts may not be effective and sustainable. Furthermore, the regulatory models of various countries are still different at the implementation level, especially since the cycle of developing a common governance framework is still very long. As a process involving the participation of multiple stakeholders, DWP governance needs to further improve the structure of the governance system from the legal and institutional levels, contributing to its own unique and effective governance system [67].

Furthermore, developed countries and regional organizations have built a complete set of laws and regulations for the implementation of EPIs, which cannot be without flaws. Laws and regulations have been formulated from the national level to the regional and river-basin levels. Various technical standards, economic policies, administrative management policies, and other detailed rules and regulations have been formulated from the technical level, management level, and social public participation level, forming a complete system. These laws and regulations have effectively promoted the implementation of EPIs [68].

In addition, the effective implementation of EPIs by multi-stakeholders depends on information coordination, which requires a strong, cohesive party, multiple cooperative subjects, long-term cooperation, and an open and transparent information-sharing platform [66,69].

## 6. Conclusions

Although there are differences in the financial democratic systems between China and the UK, China can still learn from the long and successful experience of the UK to govern the DWP. Both countries follow the three principles (the polluter pays principle, the beneficiary pays principle, and the user pays principle) to adopt similar EPIs to control water pollution, but there are obvious differences in governance goals, government supervision models, and the achievements of the economic policy governance instruments. China has a relatively higher institutional density and tends to issue control command-based policies, while the UK favors incentive-based policies. A strict top–down regulatory system cannot fully and effectively realize collaborative governance among multiple departments. At the same time, the government should implement a flexible regulatory system to give full play to the market vitality of EPIs and achieve multi-stakeholder co-governance. Moreover, the hierarchical and classified laws and regulations of each region must provide sufficient support for government governance and market governance. After all, a country as large as China does not have exactly the same governance for any region. Similarly, a transparent, efficient, and multi-party information-sharing mechanism is also indispensable. Last but not least, public welfare funds still play a small role in China's water pollution control.

## 7. Policy Implication

Although the UK's EPIs have achieved rapid development, the national conditions and systems of China and the UK are different. Some systems, such as deposit refunds, are still not feasible for China at the current stage. Therefore, it is necessary to learn from the part of the UK's policy design and implementation to make a breakthrough in China's water pollution control based on the differences in water resources, environmental resource endowment, water quality, and target management needs in various zones while paying close attention to the whole life cycle of agricultural production. This paper forms a relatively complete EPI system that provides "the whole process" incentives and constraints of DWP control from five dimensions: factor input, internalization of external costs in post-processing, market and trade, voluntary behavior, and finance and insurance (see Figure 4).

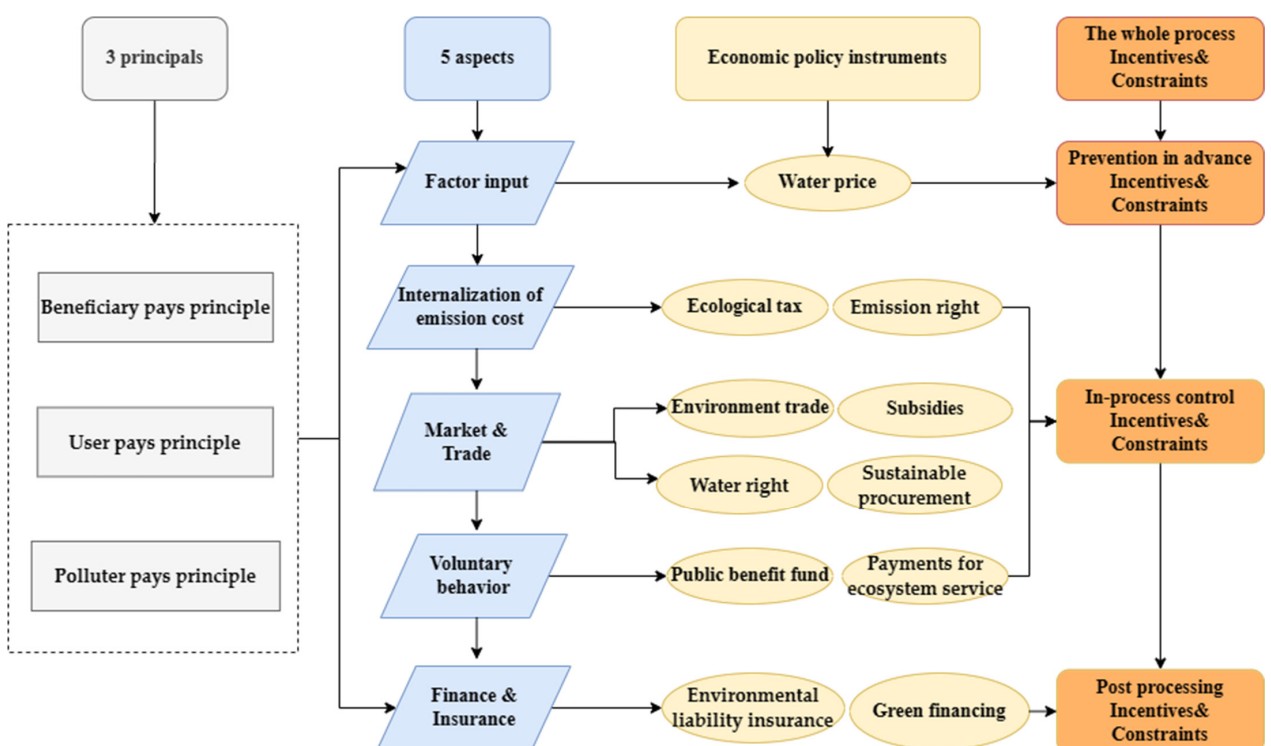

**Figure 4.** The EPI system outlook of DWP in China.

In the prevention process, the agricultural water price needs to be reformed, and each region should formulate a water price system that suits its conditions, fully considering the cost, consumption, season, and abundance of water resources, improve water use efficiency, and prevent and control DWP. At the same time, the government should give priority to social co-governance, actively exert the power of villagers' self-government associations, and cultivate farmers' green production cognition. In the in-process, cyclical subsidies should be increased appropriately due to the long-time DWP control, and the goal of subsidies should be changed to protect the environment. At least for now, certain environmental requirements should be met when applying for subsidies, and an open and transparent subsidy application environment should be provided for farmers.

Furthermore, policymakers should reform the ecological tax, the emission right trading, and the tradeable water rights system and fully consider the emission reduction targets of major water pollutants in different regions actively using the tradeable water rights, SPS, and international trade to control DWP beforehand. Each province should increase the payments for ecosystem services in rural areas, encouraging the participation of welfare funds with their efforts to establish farmers' awareness of environmental protection and form a long-term protection mechanism. In the post-processing, financial institutions should further improve the green financing mechanism and support the issues of green credit of tradeable water rights and emission rights trading. Meanwhile, governments should promote the development of the green bond market and encourage enterprises and financial institutions to issue green bonds and ELIs. In addition to that, at present, the government urgently needs to strengthen the legislative work and implementation measures of the ELI, the tradeable water rights, and the emission right trading to activate the vitality of the secondary market and break through the regional barriers to trade. Finally, the government should reform the management system of DWP and establish an efficient and cooperative governance mechanism.

**Author Contributions:** Conceptualization, J.Z., X.C. and F.L. writing—original draft preparation, J.Z., X.C. and F.L.; writing—review and editing, B.W. and N.Y.; visualization, J.Z. and M.D.; supervision, F.W.; project administration, F.W.; funding acquisition, F.W. All authors have read and agreed to the published version of the manuscript.

**Funding:** This research was funded by the National Social Science Fund of China (No. 19BGL152), China Scholarship Council (No. 201906910095), and the 2022 "Double support" Project-Innovation Team of Sichuan Agricultural University (No. 2221993021).

**Institutional Review Board Statement:** Not applicable.

**Data Availability Statement:** Not applicable.

**Acknowledgments:** We are grateful for the funding support from the National Social Science Fund of China, the China Scholarship Council, and Sichuan Agricultural University. Last but not least, we express our sincere gratitude to the two anonymous reviewers for their objective and constructive comments.

**Conflicts of Interest:** The authors declare no conflict of interest.

## Nomenclature

| | |
|---|---|
| diffuse water pollution from agriculture | DWP |
| economic policy instruments | EPIs |
| European Union | EU |
| catchment sensitive farming | CSF |
| Sustainable procurement strategy | SPS |
| Environmental liability insurance | ELI |

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
