# Peer review of "The Effects of Economic Policy Instruments of Diffuse Water Pollution from Agriculture: A Comparative Analysis of China and the UK"

_water, doi:10.3390/w15040637_

Round 1

Reviewer 1 Report

Dear Authors,

I am glad that I had the opportunity to review the manuscript entitled: "The economic policy system of diffuse water pollution from agriculture: a comparative study between China and UK" by Jinpeng Zou, Xiaodong Chen, Fang Liu, Fang Wang, Bin Wu and Ni Yang.

In this manuscript, the Authors undertook a very difficult task, which was to analyze the economic policy system of diffuse water pollution from agriculture in China. The Authors presented suggestions and countermeasures on improving China's policy and economic policy instruments system.

I consider the subject of research to be very important for the development of modern sustainable agriculture. I was very interested in the results of the considerations contained in the manuscript, and I have no doubt that they should be published.

However, I doubt whether the manuscript in its current form (as an "article") can be published. In my opinion, the Authors should explore the literature data a bit more and publish this manuscript as a "review" article. Either way, Authors should definitely add a "Conclusion" section.

The entire manuscript is carelessly adapted to the requirements of the Water journal template.

In the keywords section, please do not include words that duplicate words in the title of the manuscript. And in addition, keywords should be arranged alphabetically.

The Authors use too many abbreviations, which makes it difficult to read and understand the text. This is especially true of figures and tables, which should be understandable without having to look for explanations of abbreviations in the text.

Figure 3 is sloppy, please use a similar font size in (a) and (b).

In conclusion, I believe that the study presented in this manuscript is interesting and valuable, and the Editors of the journal Water should consider publishing it.

Reviewer 2 Report

The paper has an interesting comparative study between UK and China about the economic policy instruments (EPIs) in water pollution.

Methodologically, you need to deep understand about the comparative study. 

Charles Ragin, David Zaret, Theory and Method in Comparative Research: Two Strategies, Social Forces, Volume 61, Issue 3, March 1983, Pages 731–754, https://doi.org/10.1093/sf/61.3.731

In your case, a simple comparison between two different economic system and the   economic policy instruments (EPIs) is the main failure of your study.  UK as a more liberal economic policy with the finance democratic system has a vast difference between China. You cannot compare how environmental regulation and water quality governance organized. 

Your paper needs a strong study about the water quality governance. Without that, your analysis with Financial subsidy Subsection, Water price,. Payments for Ecosystem Services (PFS),Tradeable water rights (TWR) .... so on could not clearly originate the academic discussion. 

Recommend to read these papers, and citing and write a theoretical framework on water quality governance. 

Withanachchi, S. S., Ghambashidze, G., Kunchulia, I., Urushadze, T., & Ploeger, A. (2018). A paradigm shift in water quality governance in a transitional context: A critical study about the empowerment of local governance in Georgia. Water10(2), 98.

Wuijts, S., Driessen, P. P., & Van Rijswick, H. F. (2018). Towards more effective water quality governance: A review of social-economic, legal and ecological perspectives and their interactions. Sustainability10(4), 914.

Berry, K. A., Jackson, S., Saito, L., & Forline, L. (2018). Reconceptualising water quality governance to incorporate knowledge and values: Case studies from Australian and Brazilian Indigenous communities.

so on

Figures are not well-prepared. Please use good software to recreated them all. 

This paper does not include any discussion where the authors' contribution has contributed. There is no conclusion. It is a must in a research paper. 

The paper must be rewritten. 

Round 2

Reviewer 2 Report

Your paper ist still lacking the recommended literature background. 

Please kindly read following papers to get the strong knowledge. It is important to discuss it with finding.

Withanachchi, S.S.; Ghambashidze, G.; Kunchulia, I.; Urushadze, T.; Ploeger, A. A Paradigm Shift in Water Quality Governance in a Transitional Context: A Critical Study about the Empowerment of Local Governance in Georgia. Water 2018, 10, 98. https://doi.org/10.3390/w10020098

Try to connect the different water quality governance 

You must combine conclusion and the recommendations
